# A Candidate for Multitopic Probes for Ligand Discovery in Dynamic Combinatorial Chemistry

**DOI:** 10.3390/molecules24112166

**Published:** 2019-06-08

**Authors:** Keiko Yoneyama, Rina Suzuki, Yusuke Kuramochi, Akiharu Satake

**Affiliations:** 1Graduate School of Science, Tokyo University of Science, 1-3 Kagurazaka, Shinjuku-ku, Tokyo 162-8601, Japan; 1317828@alumni.tus.ac.jp; 2Graduate School of Chemical Sciences and Technology, Tokyo University of Science, 1-3 Kagurazaka, Shinjuku-ku, Tokyo 162-8601, Japan; B115804@alumni.tus.ac.jp; 3Department of Chemistry, Faculty of Science Division II, Tokyo University of Science, 1-3 Kagurazaka, Shinjuku-ku, Tokyo 162-8601, Japan; kuramochiy@rs.tus.ac.jp

**Keywords:** dynamic combinatorial chemistry (DCC), dynamic combinatorial library (DCL), supramolecular macrocycle, zinc porphyrin, multifunctionalized material, complementary coordination, copper(I)-catalyzed alkyne-azide cycloaddition (CuAAC), gel permeation chromatography (GPC), amphiphilic

## Abstract

Multifunctionalized materials are expected to be versatile probes to find specific interactions between a ligand and a target biomaterial. Thus, efficient methods to prepare possible combinations of the functionalities is desired. The concept of dynamic combinatorial chemistry (DCC) is ideal for the generation of any possible combination, as well as screening for target biomaterials. Here, we propose a new molecular design of multitopic probes for ligand discovery in DCC. We synthesized a new Gable Porphyrin, **GP1**, having prop-2-yne groups as a scaffold to introduce various functional groups. **GP1** is a bis(imidazolylporphyrinatozinc) compound connected through a 1,3-phenylene moiety, and it gives macrocycles spontaneously and quantitatively by strong imidazole-to-zinc complementary coordination. Some different types of functional groups were introduced into **GP1** in high yields. Formation of heterogeneous macrocycles composed of GP1 derivatives having different types of substituents was accomplished under equilibrium conditions. These results promise that enormous numbers of macrocycles having various functional groups can be provided when the kinds of GP components increase. These features are desirable for DCC, and the present system using **GP1** is a potential candidate to provide a dynamic combinatorial library of multitopic probes to discover specific interactions between a ligand and a biomaterial.

## 1. Introduction

Ligand discovery for target proteins is an important issue at the initial stage in pharmaceutical research. To discover the proper ligand, not only molecular modeling simulations based on the crystal structures of target proteins, but also practical binding experiments using large and diverse compound collections are required. The concept of dynamic combinatorial chemistry (DCC) [1,2,3,4,5,6] is significant for efficient screening to find interactions between a ligand and a target biomaterial [7,8,9,10,11,12], such as proteins and nucleic acids, and in fact, DCC is now becoming a powerful tool for drug discovery [13].

Multifunctionality is necessary to recognize the target cell followed by ligand discovery for proteins in the target cell. In such cases, a combination of more than two keys recognizing the target cell becomes important as well as interaction with the protein. However, rising numbers of components exponentially increase their combinations in the multicomposites. Therefore, innovative multitopic DCC methods that are more combinatorial, more diverse, and more dynamic on their multitopic sites should be developed. In Figure 1, an idea of advanced multitopic DCC systems is shown. A macrocycle is composed of six ditopic components, which are self-assembled to each other by exchangeable coordination bonds. When five kinds of self-assembled macrocycles (symbolized as five colors) are mixed, and recombination of each component occurs among the macrocycles, various heterogeneous macrocycles will be generated randomly as shown by multiple colored rings. A structurally reliable macrocyclic motif is beneficial for surveying a combination of components for targets. On the application of the macrocyclic system into DCC, highly combinatorial, highly diverse, and highly dynamic features are expected.

So far, several multifunctionalized materials have been reported. Dendritic polymers [14,15], mesoporous-silica-based materials [16], graphene oxides [17], carbon dots [18], and carbon nano-onions [19] are proposed for versatile materials. However, most of their combinatorial systems are static, not dynamic. Only limited cyclic pseudopeptidic compounds connected through disulfide bonds are reported for DCC [20]. In a static combinatorial library, the numbers of combinations to be tried within all of the possible combinations are limited due to time and resources constraints. On the other hand, DCC using a multitopic system is expected to produce every possible combination and enormous multifunctionalized material collections.

In general, DCC requires quantitative and reversible reactions or interactions to provide diverse composite collections. The exchange rates should be fast to generate all the possible combinations of all of the components within a realistic timescale. Furthermore, a composite interacted with a target biomaterial must be stable enough to detect and isolate the target. For high-throughput separation and detection, liquid chromatography (LC) with characteristic and sensitive UV–vis absorption and/or fluorescence monitoring is one of the most suitable methods, and the subsequent mass spectrometric analysis of the isolated composite/target fraction provides information about components in the composite.

We have continued with the idea that imidazolyl zinc porphyrin (ImZnP)-based self-assembled nanorings are possible candidates to provide the above DCC system. The ImZnP system was initially developed by Kobuke and Miyaji as a biomimetic model of the so-called “special pair” in photosynthetic reaction center of photosynthetic purple bacteria [21] (Figure 2). The slipped-cofacial dimer of the imidazolyl zinc porphyrins mimics arrangement of the bacteriochlorophylls of the special pair. Characteristic features of ImZnP are as follows: (1) reversible and strong complementary imidazole-to-zinc coordination bonds, reaching a self-assembled constant, *K*_a_ > 10^8^ M^−1^; (2) easily detectable and distinguishable split Soret bands in a range between 410 and 430 nm with large molar extinction coefficients; (3) fluorescent, even in their self-assembled states. These features are considered to be desirable to apply the system to DCC.

We have developed ImZnP into bis-ImZnP systems, in which two ImZnPs are connected directly or through appropriate spacers. They spontaneously produce oligomeric/polymeric and cyclic [22,23]/linear array systems [24,25,26]. For example, a Gable zinc porphyrin (GP), in which two ImZnPs are connected through an *m*-phenylene moiety, gave a mixture of pentameric and hexameric GPs quantitatively in solution (Figure 3 and Figure 4).

In our previous work, chloroform was mostly used to treat ImZnP and bis-ImZnP systems, because it is a “good” solvent for solvation of the π-aromatic skeleton of porphyrins, while not interfering with the coordination between the imidazolyl moiety and the zinc ion. In the presence of coordinating solvent, such as methanol, exchange rates among imidazolyl and zinc moieties were accelerated, and their compositions varied depending on different concentrations and temperatures. Therefore, the system is concluded to be a dynamic one. In our previous molecules, it was difficult to treat ImZnP systems in polar solvents. For example, in acetonitrile and water, ImZnP systems tend to give undesirable aggregates, which can be monitored by UV–vis absorption spectra as deviating from the Lambert–Beer law and light scattering. To apply ImZnP systems to DCC in biorelated chemistry, the hydrophilicity of the ImZnP moiety must be increased to dissolve them in aqueous solution, and also scaffolds to introduce arbitrary substituents should be attached. Here, we present a new GP having hydrophilic side chains on the four *meso*-positions and prop-2-yne groups on the imidazolyl groups as scaffolds as shown in Figure 5. In the new system using **GP1**, we should verify whether (1) complementary imidazolyl-to-zinc coordination occurs even in a polar solvent system, (2) various substituents can be introduced into the prop-2-yne groups, and (3) dynamic exchange occurs between different GPs to give heterogeneous self-assembled rings.

In general, GP gives a mixture of pentameric and hexameric rings, in which the combinations of coordination patterns are random, as shown in Figure 6. Thus, besides the five-fold symmetry structure (Out–In/Out–Out/In–In = 5/0/0, clockwise), the pentamer can take another three asymmetric structures (two 3/1/1s and 1/2/2) as shown in Figure 6. In the case of the cyclic hexamer, eight possible coordination isomers are considered (Figure 6). Because the rates of rotation about carbon–carbon bonds between porphyrin and the adjacent *m*-phenylene moiety are slow, they are distinguished in ^1^H NMR spectra, giving complicated signal patterns by overlapping those of conformational isomers. Assignment of each signal in the ^1^H-NMR chart is difficult by using standard two-dimensional methods on a 500 MHz NMR spectrometer. In the case of simpler trisporphyrin (TrisPor) systems [27], only two conformational isomers are possible (Figure 7). Thus, the ^1^H-NMR spectrum is simpler than those of pentamers and hexamers composed of GPs, and, in fact, parts of signals can be assigned as each coordination isomer. In the case of bis-zinc porphyrin connected through 1,3-bisethynylbenzene, a mixture of self-assembled pentamer and hexamer are also produced similar to the case of GP [28]. In this system, however, the ^1^H-NMR spectra are very simple due to the fast rotation about the C–C bonds between the ethynylene moieties and their adjacent substituents. Therefore, using GPs is beneficial to produce various conformational isomers, and this nature is rather convenient to prepare a large and diverse dynamic combinatorial library (DCL).

Although ^1^H-NMR spectroscopy is sensitive to distinguishing the conformational isomers, UV–vis spectroscopy is insensitive for the difference among the conformational isomers. UV–vis spectroscopy can distinguish only the difference between pentameric and hexameric rings, but not their conformational isomers. In the case of hexameric rings, the peak maximum of one of the split Soret bands appears on the longer wavelength side by approximately 2 nm, compared with that in corresponding pentameric rings [29]. This feature is convenient to simplify the analysis of gel-permeation chromatography (GPC) to only discriminate pentamer from hexamer using a photodiode array (PDA) detector.

Another advantage of using porphyrin derivatives is that they are easily detectable by matrix-assisted laser desorption ionization time-of-flight mass spectrometry (MALDI-TOF MS). Although pentameric and hexameric rings are stable in solution when conducting GPC analysis, they are dominantly detected as the monomeric component of GP under high vacuum MALDI-TOF MS conditions. This nature will be enough to analyze the components of heterogeneous composites. Therefore, high-throughput analysis is expected by a combination of LC and MALDI-TOF MS for ligand discovery in DCC using this system.

In this paper, we describe the synthesis of new **GP1** and the introduction of various substituents on **GP1** to give four derivatives. On the basis of the expected properties of GP systems, their dynamic behaviors are examined.

## 2. Results and Discussion

### 2.1. Synthesis of Porphyrins

Synthesis of **GP1** was performed as shown in Scheme 1 and Scheme 2. Condensation of two kinds of aldehydes **2** and **3**, and dipyrromethane **4** in the presence of trifluoroacetic acid (TFA), followed by subsequent oxidation with *p*-chloranil gave a mixture of porphyrins **5**, **7**, and **8**. After purification of monoimidazolyl porphyrin **5**, the acetal group was deprotected to give aldehyde **6**.

Aldehyde **6** and dipyrromethane **4** were reacted in the presence of TFA, and subsequent addition of aldehyde **2** to the mixture gave a mixture of bisimidazole monoporphyrin **8**, trisporphyrin **9**, and the free-base form of target gable porphyrin **GP1Fb**. The free-base **GP1Fb** was isolated by preparative recycle GPC systems (Appendix A).

Introduction of zinc ion into free-base porphyrin **5** and **GP1Fb** was performed, respectively, as shown in Scheme 3 and Scheme 4. Monoimidazolyl zincporphyrin **5Zn** gave the complementary coordination dimer spontaneously in chloroform. In Figure 8, the ^1^H-NMR spectra of free-base porphyrin **5** and coordination dimer of zinc porphyrin **5Zn** are shown. Characteristic signals observed at 5.82, 5.41, and 2.19 (and 2.15) in the dimer are assigned as *Im*_5_, *β*_1_, and *Im*_4_ (distinguished as tautomers), respectively, which are significantly shielded by the facing porphyrin in the dimer. The coordination dimer of zinc porphyrin **5Zn** includes their tautomers concerning aryl groups, and the parts of signals are distinguished in the ^1^H-NMR spectrum.

In the case of **GP1Fb**, introduction of zinc ions can be confirmed by UV–vis spectrum in pyridine, in which complementary coordination bonds are disconnected by competitive coordination of pyridine, and MALDI-TOF MS. The ^1^H-NMR spectrum of **GP1** in CDCl_3_ was too complicated to be assigned due to the formation of various self-assembled oligomers including their tautomers. However, characteristic shielded signals, *β*_1_, *Im*_4_, and *Im*_5_, are observed in the range of 6.00–4.51 (*β*_1_, *Im*_5_) and 2.10–1.60 (*Im*_4_) (Figure 9). These results show that **GP1** forms supramolecular structures by the complementary imidazole-to-zinc coordination bonds.

### 2.2. Introduction of Substituents

As a preliminary study, some different types of substituents, hydrophilic oligoether (**PEG8**), pyrene (**PY**) as a large aromatic compound, a branched alkyl group (**BA**) derived from FINEOXOCOL^®^ 180, and 3,3,4,4,5,5,6,6,6-nonafluorohexyl (**F9**) were planned to be introduced into **GP1** using copper(I)-catalyzed alkyne-azide cycloaddition (CuAAC) (Scheme 5). Before applying the reaction to **GP1**, we have examined experimental conditions on **5Zn** (Scheme 6). Because a free-base porphyrin reacts with copper ion to give a corresponding copper porphyrin quantitatively, from which it is difficult to remove the copper ion, zinc porphyrin derivatives must be used as substrates of CuAAC, and amounts of the copper ion should be limited to prevent metal exchange reaction from zinc to copper on the porphyrin. Finally, we found the following appropriate conditions: in the presence of ca. six equiv of CuI and ca. 17 equiv of 2,6-lutidine, at rt, for more than 7 days. Under these conditions, CuAAC proceeds in high yield without transmetalation even in various solvents. The reaction progress was monitored using thin layer chromatography (TLC) and MALDI-TOF MS by detecting target signals directly and the following purification was carried out by silica-gel column chromatography followed by reprecipitation from a mixture of chloroform and hexane. The ^1^H-NMR spectrum of **PEG8-5Zn** in CDCl_3_ is shown in Figure 10. After substitution of the PEG8 group, signals on the porphyrin skeleton are similar between **5Zn** and **PEG8-5Zn** and extra signals corresponding to the newly introduced PEG8 group are mainly observed in the range of 4.0–3.0 and 1.7–1.0 ppm. Parts of the extra signals are shifted upfield, suggesting that they are located over the porphyrin planes.

In a similar procedure, **PY** (pyrene) and **F9** groups were introduced into **5Zn** to give **PY-5Zn** and **F9-5Zn** in 95% and 88% yields, respectively. The CuAAC reaction condition was confirmed to be appropriate to introduce various functional groups into **5Zn**. In the case of **GP1**, initially, we were afraid that efficient CuAAC might not proceed very well because **GP1** had two reaction parts of alkyne groups, which might be sterically hindered by formation of its self-assembled macrocycles. Against our anxiety, CuAAC also proceeded on self-assembled structures composed of **GP1** in good yields under similar conditions. Their bis-substituted structures and purities as porphyrin derivatives were confirmed by MALDI-TOF MS and HPLC-GPC systems equipped with a PDA detector using pyridine as an eluent, where dissociated GP units can be observed. In Figure 11, MALDI-TOF MS spectra of **GP1**, **PEG8-GP1**, **PY-GP1**, **BA-GP1**, and **F9-GP1** with DCTB (*trans*-2-[3-(4-*tert*-butylphenyl)-2-methyl-2-propenylidene]malononitrile) as a matrix recorded on JSM-S3000 (JEOL) with spiral mode are shown. Each molecular weight expected as a target compound is shown in Scheme 5. On the MALDI-TOF MS, signals were observed as sodium ion adducts [M + Na]^+^. From the highly resolved spectra, these derivatives can be easily assigned and distinguished from others. In the MALDI-TOF MS analysis, only signals corresponding to components of GPs are detected, even though they existed as cyclic pentamers and hexamers in solution. This property is not inadequate, rather it is convenient to analyze the components of heterogeneous self-assembled cyclic oligomers composed of different components as described in the later part.

### 2.3. UV–Vis Absorption and Fluorescence Spectra in Aqueous Solution

To examine the application of our new porphyrin system in aqueous media, UV–vis absorption and fluorescence spectra of **5Zn** and **PEG8-5Zn** were recorded in mixtures of different compositions of water and acetonitrile (Figure 12 and Figure 13). In general, random aggregation of porphyrin causes absorption spectral broadening and fluorescence quenching. In the case of nonsubstituted **5Zn**, the fluorescence quantum yields were almost constant between 100% acetonitrile and up to 60 vol% of water in a mixture with acetonitrile, whereas the fluorescent nature was decreased significantly above 60 vol% of water solutions (Figure 12c). UV–vis absorption spectrum of **5Zn** in the 80 vol% water solution became broader, showing higher aggregated states of the dimer of **5Zn** (Figure 12b). On the other hand, in the case of **PEG8-5Zn**, the fluorescence quantum yields were almost constant between 100% acetonitrile and up to 90 vol% of water solution, and even in 100% water solution, half of the intrinsic fluorescent property remained (Figure 13c). The UV–vis absorption spectrum in the 100% water solution did not show remarkable aggregated structure (Figure 13b). Therefore, the introduction of a hydrophilic PEG8 group over the porphyrin plane is effective in preventing the production of higher aggregated structure. Unfortunately, **PEG8-GP 1** was difficult to dissolve in 100% water. Therefore, further hydrophilic substituents should be introduced into GPs when they are applied to biorelated materials. Longer polyethylene glycols and charged compounds are candidates to improve the solubility of GP derivatives. GPs have not only characteristic absorption bands in the visible region with high extinction coefficients, but also an intrinsic fluorescence property to emit red light as well as **5Zn**. If the solubility of GP derivatives increases under physiological conditions, the fluorescent feature will be useful to be detected as a super complex with target materials.

### 2.4. Reconstitution of GPs

In order to verify whether our self-assembled porphyrin ring systems can produce heterogeneous composites, reconstituted porphyrin rings composed of two components were compared with those of sole components. The comparisons were performed on improvement of solubility in polar acetonitrile, changes of elution curves in GPC analyses, and their NMR spectra.

At first, reorganized porphyrin rings composed of sole components were prepared. Reorganization procedures to converge into a mixture of pentamer and hexamer were modified from the previous method [29]. **GP1**, **PEG8-GP1**, **PY-GP1**, **BA-GP1**, and **F9-GP1** were independently reorganized in a 20 µM solution of a mixture of chloroform and methanol (7:3) at rt for 12 h. All of the GPs dissolved well in chloroform. After the reorganization, the mixture was washed with water to remove methanol, and filtrated the residual water on phase separator paper (Whatman^®^) to give methanol-free chloroform solution. In the absence of methanol, the reorganization is suppressed kinetically. In Figure 14, GPC charts of before and after the reorganization of **PEG8-GP1** are shown. Before the reorganization, GPs formed various acyclic oligomeric structures accompanied by target cyclic pentamers and hexamers. The acyclic oligomers tend to be absorbed on the polystyrene gel in GPC columns due to presence of their terminal imidazolyl moieties to cause the heavily tailing charts. After the reorganization, the free-imidazolyl moieties are disappeared by convergence into cyclic structures to give converged elution curves without tailing.

In Figure 15, GPC charts of reorganized **GP1**, **PY-GP1**, and **PEG8-GP1** are arranged with the same range of retention time (RT). In these analyses, two columns were connected to extend the analytical length of GPC. Because these elution curves are almost reproduced in several reorganization experiments using the same conditions, the elution curves are considered to show product distributions after equilibria among pentamers and hexamers of the GPs.

Figure 16 shows the UV–vis absorption spectra recorded at the time range of 16–20 min on the PDA detector in the GPC experiments (Figure 15c). In Figure 16a, the spectra before the peak maximum (17.8 min) in Figure 15c are overlaid, whereas those after the peak maximum are arranged in Figure 16b. Therefore, the changes are uphill and downhill roads in (a) and (b), respectively. On the uphill road in (a), the wavelengths of the peak maxima around 448 nm seem to be constant at 448 nm, indicating that the components are dominantly hexamers. On the downhill road in (b), however, those seem to be shifted from 448 to 446 nm around the top part. This observation in (b) indicates that contents of elution varied from hexamers to pentamers at this point, and the pentamers are eluted at later RT than the hexamers. The behavior that smaller molecules elute at later RT is consistent with the characteristics of the GPC column.

Distributions of hexamer and pentamer composed of **PEG8-GP1** varied under different temperatures in acetonitrile solution. Figure 17 shows GPC charts of the mixtures reorganized at 0, 25, 40, and 60 °C. The relative ratio of pentamer increased upon raising the reorganization temperature, indicating that the formation of the smaller ring, namely the pentamer, is entropically favorable. These results clearly show that the present system relies on equilibrium, which responds to external environments.

To prepare heterogeneous composites, an equivalent of **PY-GP1** dissolved in the minimum amount of chloroform was added to an acetonitrile/methanol solution (2:1 *v*/*v*) of the reorganized ring of **PEG8-GP1**, and the mixture was stirred for 12 h at 60 °C. During the reconstitution process, no precipitate was formed, and it was confirmed by the UV–vis spectrum that all porphyrin derivatives were recovered in the acetonitrile solution after the reconstitution. In Figure 18, GPC charts of the samples at each step are arranged in the same RT scale. The shape of an as-prepared mixed sample (c in Figure 18) seemed to be already changed from the simple summation of individual samples (a: **PY-GP1** and b: **PEG8-GP1** in Figure 18). After reconstitution, the distribution looks like a simple Gaussian shape as shown in Figure 18d. Because such an apparently simple distribution has never been observed in the case of reorganization in single GPs, this phenomenon must be due to reconstitution of **PY-GP1** and **PEG8-GP1**. Overlaid spectra recorded on the PDA detector in the GPC analysis shown in Figure 18d are represented in Figure 19. Interestingly, peak maxima of the Soret band at the longer wavelength seem to be constant at 446 nm, suggesting the dominant formation of pentamer. Although porphyrin parts were almost constant in the normalized UV–vis absorption spectra (Figure 20), which were obtained from the overlaid spectra in Figure 19 with normalization at 446 nm, relative intensities of pyrene parts (250–350 nm) were increased in the later RT region. By using molar extinction coefficients of pyrene derivatives and porphyrin derivative, the ratio of **PY-GP1** and **PEG8-GP1** can be estimated approximately as 2:3 at RT 17.70 min, and 3:2 at RT 18.18 min. Therefore, even in the apparently simple distribution, the constitutions of **PY-GP1** and **PEG8-GP1** vary continuously.

^1^H-NMR spectra of **PY-5Zn**, **PEG8-GP1**, **PY-GP1**, and the reconstituted sample of **PEG8-GP1** and **PY-GP1** (1:1) are shown in Appendix A. From the assigned signals corresponding to the shielded pyrene protons in **PY-5Zn**, signals observed in the region of 8.0–7.2 ppm can be estimated as parts of pyrene protons existing outside the pentameric rings. Because the relative integration ratio of the region 8.0–7.2 ppm of the reconstituted sample decreased compared with those of sole rings composed of **PY-GP1**, pyrene moieties existing outside the porphyrin ring decreased, suggesting that pyrene moieties existing inside the porphyrin ring become dominant. The pyrene protons inside the ring are considered to be more shielded by other pyrenes and porphyrins, and they are probably shifted upfield.

Because **PY-GP1** was less soluble in the polar mixed solution, whereas **PEG8-GP1** was well soluble, we expected the formation of a 3:3 micelle-like cyclic hexamer, in which hydrophilic PEG8 groups were arranged outside and hydrophobic PY groups were gathered inside the cyclic hexamer, as shown in Figure 21a. In such a structure, π–π interactions among PY moieties as well as PY and zinc porphyrin moieties were expected, and it might stabilize the structure. However, the result was not as expected, and only the cyclic pentamer was observed as a mixture of conformational isomers as shown in Figure 21b, in which, at least one of the pyrene moieties was exposed to the polar solvent on the outside of the ring. This result suggests that entropic gain to prepare a large number of smaller macrocycles, pentamers, instead of hexamers, overcomes the destabilization energy to expose pyrene moieties to polar solvent.

By similar procedures, reconstitution of **BA-GP1** and **F9-GP1** with **PEG8-GP1** was performed. In both cases, all of the porphyrin derivatives were recovered quantitatively in acetonitrile solution without precipitation. GPC-PDA analyses of the mixtures reconstituted at various temperatures are shown in Appendix A. In these cases, mixtures of hexamers and pentamers were obtained, and their relative ratios seem to be almost constant at the various temperatures. Although their ^1^H-NMR spectra were too complicated to analyze and discuss their structures (Appendix A), ^19^F-NMR spectral analysis made it easy. ^19^F-NMR spectra of **F9-N3**, **F9-5Zn**, **F9-GP1**, and a reconstituted sample of **F9-GP1** and **PEG8-GP1** are shown in Figure 22, Figure 23 and Figure 24.

In Figure 22a, four kinds of nonequivalent fluorine signals of **F9-N3** were observed at –80.9, –114.0, −124.4, and −125.9 ppm, which were assigned by referring reported values [30] as 6-, 3-, 4-, and 5-positions, respectively. ^19^F-NMR signals of **F9-5Zn**, −81.0, −114.8, −124.6, and −126.1 ppm, seem almost the same, suggesting that the effects of the porphyrin are very small in the dimer. ^19^F-NMR spectra of **F9-GP1** before and after reorganization are shown in Figure 23a,b, respectively. In Figure 23a, the four kinds of nonequivalent fluorine atoms are split, and each lower-field signal, –80.9, –114.7, –124.5, and –126.0 is almost identical to those of **F9-N3** and **F9-5Zn**, whereas the existence of a set of the other upper-field signals, −81.2, −114.9, −124.7, and −126.3 ppm, indicates the existence of different circumstances. After the reorganization of **F9-GP1**, chemical shift values corresponding to the upper-field peaks were expanded widely in a range of approximately 400 Hz (ca 1–2 ppm). The characteristic upper-field peaks originate from fluorine atoms existing inside porphyrin rings, and their complexity indicates the existence of various conformers. On the other hand, the lower-field signals observed as singlets are assigned as fluorine atoms existing outside the porphyrin rings. As shown in Figure 24, the integration ratios of ^19^F-NMR signals originating outside and inside were changed in a reconstituted sample composed of **F9-GP1** and **TEG8-GP1**. Thus, the inside signals were increased by comparing with the outside, suggesting that fluorinated groups in **F9-GP1** compounds tend relatively to exist inside the ring. These data also show the formation of heterogeneous composites composed of **F9-GP1** and **PEG8-GP1**.

Formation of the heterogeneous self-assembled macrocycles composed of **F9-GP1** and **PEG8-GP1**, as well as **PY-GP1** and **PEG8-GP1**, suggests that any kind of functional groups that are even intrinsically phobic to one another, can be blended through the macrocycles. This is because the formation energy of imidazolyl-to-zinc complementary coordination is much larger than those of interactions among the substituents. Thus, the substituents do not interfere with the formation of the macrocycles. We consider that statistically heterogeneous macrocycles will be obtained when larger numbers of GP components are mixed. This feature is suitable for DCC.

## 3. Materials and Methods

**General Procedure.** All chemicals and solvents were of commercial reagent quality and used without further purification unless otherwise stated. 3-(4,4-Dimethyl-2,6-dioxan-1-yl)benzaldehyde **3** [29] and 2,2′-(2,5,8,11-tetraoxapentadecane-15,15-diyl)bis(1*H*-pyrrole) **4** [31] were prepared according to the literature. PEG8N_3_ (25-Azido-2,5,8,11,14,17,20,23-octaoxapentacosane) was purchased from TCI (Tokyo, Japan). CHCl_3_ (Kanto, extra pure) stabilized with 0.5–1% ethanol was used. Reactions were monitored on silica gel 60F_254_ TLC plates (Merck, Tokyo, Japan). Silica-gels utilized for column chromatography were purchased from Kanto Chemical Co. Inc. (Tokyo, Japan): Silica-Gel 60N (Spherical, Neutral) 63–210 μm and 40–50 μm (Flash). ^1^H-NMR spectra were recorded by using JEOL ECA-500 (500 MHz, JEOL, Tokyo, Japan), JEOL ECA-300 (300 MHz, JEOL, Tokyo, Japan ), or JEOL ECZ-400 (400 MHz, JEOL, Tokyo, Japan ) and chemical shifts were recorded in parts per million (ppm) relative to tetramethylsilane. High-resolution MALDI–TOF mass spectra were collected on JEOL JMS-S3000 SpiralTOF with dithranol or *trans*-2-[3-(4-*tert*-butylphenyl)-2-methyl-2-propenylidene]-malononitrile (DCTB) as a matrix containing sodium iodide (NaI) and polyethylene glycol as an internal or an outer standard. The data analyses were carried out on mMass ver. 5.5 (http://www.mmass.org/). UV–vis absorption spectra were collected on JASCO V-650 or V-660 spectrometer (JASCO Co. Tokyo, Japan). Steady-state fluorescence spectra were collected on Hitachi F-4500 spectrometer (Hitachi High-Tech Science Co., Tokyo, Japan) and corrected for the response of the detector system. The UV-vis absorption and fluorescence spectra were measured using a square cell (optical path = 10 mm).

Analytical high-performance liquid chromatography (HPLCs, JASCO Co. Tokyo, Japan) was carried out by using the following three systems:

**[System 1]** JASCO PU-2080plus and MD-2018plus system equipped with two TSK G2500H_HR_ (Tosoh, 7.8 mm × 30 cm, exclusion limit: 20,000 Da) and one TSK G2000H_HR_ (Tosoh, 7.8 mm × 30 cm, exclusion limit: 10,000 Da) columns using pyridine as an eluent.

**[System 2]** JASCO PU-2089 and MD-44010 system equipped with two TSK G4000H_HR_ (Tosoh, 7.8 mm × 30 cm, exclusion limit: 40,000 Da) columns using CHCl_3_/THF (95:5 *v*/*v*) as an eluent.

**[System 3]** JASCO PU-2089 and MD-44010 system equipped with a TSK gel α-M (Tosoh, 7.8 mm × 30 cm, exclusion limit: 10,000,000 Da) column using a mixture of water/acetonitrile (4:6 *v*/*v*) as an eluent.

Preparative gel permeation chromatography (GPC) were carried out on LC-908 (Japan Analytical Industry, Tokyo, Japan) attached to one TSK G2500H_HR_ (Tosoh, 21.5 mm × 30 cm, exclusion limit: 20,000 Da) and one G2000H_HR_ (Tosoh, 21.5 mm × 30 cm, exclusion limit: 10,000 Da) columns eluted with pyridine. Infrared (IR) spectra were measured with JASCO FT/IR-4600 (JASCO Co. Tokyo, Japan) and ATR PRO ONE (JASCO Co. Tokyo, Japan) using ATR method.

*1-(Prop-2-yn-1-yl)-1H-imidazole-2-carbaldehyde* (**2**). In a 20 mL flask were placed 1*H*-Imidazole-2-carbaldehyde [32] (50mg, 0.52 mol), LiBr∙H_2_O (300 mg, 2.86 mmol), dry DMF (0.5 mL), and propargyl bromide (1.5 equiv, 58.7 μL, 0.78 mmol). The reaction mixture was heated to 60 °C. The reaction progress was monitored with TLC and additional propargyl bromide (1.5 equive, 58.7 μL, 0.78 mmol) was added to the reaction mixture (total ten equiv). The reaction progress almost stopped after 10.5 h, and then the mixture was quenched and neutralized with sat. NaHCO_3_ aqueous solution. The product was extracted with ethyl acetate and washed with sat. NaHCO_3_ aqueous solution and brine. The organic layer was dried over anhydrous MgSO_4_ and the solvent was evaporated. Remaining DMF was removed by adding toluene using azeotropy. The residue was dried in vacuo to afford brown liquid as the crude product (35 mg). Further purification with a silica gel column (φ 3 cm × 7 cm, hexane → hexane:ethyl acetate = 1:1) gave the pure titled product as pale yellow liquid (20 mg, 29%). TLC *R_f_* = 0.6 (silica gel, chloroform:methanol = 11:1). ^1^H-NMR (300 MHz, DMSO-*d*_6_) *δ* (ppm) 9.71 (s, 1H), 7.73 (s, 1H), 7.31 (s, 1H), 5.25 (d, *J*=3.0 Hz, 2H), 3.51 (t, *J*=3.0 Hz, 1H).

*PA(PorphyrinA)* (**5**). A 500 mL three-necked flask was charged with 1-(prop-2-yn-1-yl)-1*H*-imidazole-2-carbaldehyde (**2**) (206 mg, 1.53 mmol), 3-(4,4-dimethyl-2,6-dioxan-1-yl)benzaldehyde (**3**) (365 mg, 1.53 mmol), 2,2′-(2,5,8,11-tetraoxapentadecane-15,15-diyl)bis(1*H*-pyrrole) (**4**) (1.07 g, 3.04 mmol), and chloroform (284 mL). After bubbling with N_2_ gas for 30 min, trifluoroacetic acid (TFA) (234 μL, 3.06 mmol) in chloroform was added. After stirred for 4 h under dark, the color of the reaction mixture was changed from yellow to red and spots showing red emission by irradiation with 365 nm appeared on TLC. Triethylamine (TEA) (469 μL, 3.37 mmol) was added to neutralize the solution, and the mixture was stirred until the color became orange. To the mixture *p*-chloranil (1.28 g, 5.20 mmol) was added, and the reaction mixture was stirred for 14 h. The solvent was evaporated to dryness to give a black solid (4.12 g). The crude product was purified by an alumina column (φ 8 cm × 12 cm, chloroform → chloroform:acetone = 10:1). The second band eluted with chloroform and acetone was collected. The fractions were concentrated, and the residue was further purified with a flush silica gel column (φ 3 cm × 20 cm, chloroform:methanol = 50:1). The red band was collected, and concentrated under reduced preessure to afford **5** as a purple solid (22.7 mg, 12%). TLC *R_f_* = 0.48 (silica gel, ethyl acetate); MALDI-TOF MS (matrix: dithranol) found *m*/*z* ([M + Na]^+^) 1035.5180, calcd for [C_58_H_72_N_6_O_10_ + Na]^+^ 1035.5208; ^1^H-NMR (300 MHz, CDCl_3_) *δ* (ppm) 9.55 (d, *J* = 3.9 Hz, 2H, β), 9.47 (d, *J* = 4.8 Hz, 2H, β), 8.86(d, *J* = 3.9 Hz, 2H, β), 8.78 (d, *J* = 4.8 Hz, 2H, β), 8.35, 8.26 (each s, 0.5H × 2, *Ph_2_*), 8.21 (d, *J* = 7.8 Hz, 0.5H, Ph_6_), 8.12 (d, *J* = 7.5 Hz, 0.5H, *Ph_6_*), 8.00 (d, *J* = 7.8 Hz, 1H, *Ph_4_*), 7.79–7.75 (m, 1H, *Ph_5_*), 7.73 (s, 2H, *Im_5_*), 5.67 (d, *J* = 5.4 Hz, 1H, *acetal-CH*), 5.07(t, *J* = 7.2 Hz, 4H, *TEG*), 4.38(dd, *J* = 2.7 Hz, 2H, *propargyl CH_2_*), 3.86–3.40(m, 28H, *TEG*, *acetal-CH_2_*), 3.31(s, 6H, *TEG-CH_3_*), 2.81–2.72(m, 4H, *TEG*), 2.25 (d, *J* = 2.7 Hz, 1H, *propargyl CH*), 1.36, 1.34(each s, 3H, *acetal-CH_3_*), 0.80, 0.79(each s, 3H, *acetal-CH_3_*), −2.67(s, 2H, *H_inner_*); ^13^C-NMR (75 MHz, CDCl_3_) *δ* (ppm) 148.17, 149–144, 142.46, 137.14, 137.08, 134.98, 132.56, 132.43, 132.38, 130.54, 129.48, 127.98, 126.84, 126.78, 125.82, 120.80, 119.75, 119.69, 103.19, 102.15, 77.96, 77.41, 77.20, 74.44, 72.04, 70.91, 70.88, 70.82, 70.65, 70.47, 70.10, 59.12, 37.79, 37.19, 31.32, 30.47, 23.29, 22.03.

*PA-Zn(PorphyrinA-Zn) (5Zn)*. **5** (24 mg, 2.34 × 10^−5^ mol) was dissolved in chloroform (5 mL) in a 35 mL flask. A saturated methanol solution of Zn(OAc)_2_ (17 mg, 9.4 × 10^−5^ mol) was added to the solution, and the mixture was stirred for 2 h at rt in the dark. The mixture was washed with water (×3), brine (×1) and dried over anhydrous Na_2_SO_4_. The solvent was evaporated to dryness, giving the titled compound as a purple solid (23.2 mg, 92%). MALDI-TOF-MS (matrix: DCTB) found *m*/*z* [M + Na]^+^ 1097.4863, calcd for [C_58_H_70_N_6_O_10_Zn + Na]^+^ 1097.4343; ^1^H-NMR (300 MHz, CDCl_3_, observed as atropisomers) *δ* (ppm) 9.63 (d, *J* = 4.5 Hz, 4H, β), 9.03 (d, *J* = 4.5 Hz, 4H, β), 8.93 (d, *J* = 4.5 Hz, 4H, β), 8.80(s, 1H, *Ph_2_*), 8.65 (d, *J* = 7.2 Hz, 1H, *Ph_6_*), 8.27 (s, 1H, *Ph_2_*), 8.13 (d, *J* = 7.5 Hz, 1H, *Ph_6_*), 8.08 (t, *J* = 7.5 Hz, 2H, *Ph_4_*), 7.98 (t, *J* = 7.5 Hz, 1H, *Ph_5_*), 7.80 (t, *J* = 7.5 Hz, 1H, *Ph_5_*), 5.93, 5.71(each s, 2H, *acetal-CH*), 5.83-5.81 (m, 2H, *Im_5_*), 5.41(t, *J* = 4.5 Hz, 4H, β), 5.21 (t, *J* = 6.6 Hz, 8H, *TEG*), 4.05–3.46 (m, 56H, *TEG*, *acetal-CH_2_*), 3.30 (s, 12H, *TEG-CH_3_*), 3.08–2.95 (m, 8H, *TEG*), 2.57-2.53 (m, 4H, *propargyl CH_2_*), 2.19 (s, 1H, *Im_4_*), 2.15 (s, 1H, *Im_4_*), 1.60–1.58 (m, 2H, *propargyl CH*), 1.53, 1.39 (each s, 6H, *acetal-CH_3_*), 0.95, 0.83 (each s, 6H, *acetal-CH_3_*); ^13^C-NMR (125 MHz, CDCl_3_) δ(ppm) 151.14, 149.83, 149.08, 147.82, 145.49, 143.89, 136.66, 136.45, 135.39, 132.33, 129.31, 126.41, 122, 128.36, 126.90, 124.92, 121.27, 119.23, 116.10, 102.41, 94.76, 77.97, 77.79, 75.35, 73.69, 71.81, 70.85, 70.80, 70.71, 70.63, 70.44, 70.33, 38.29, 31.82, 58.92, 58.87, 35.20, 30.44, 30.28, 23.35, 21.91.

*PA’(PorphyrinA’)* (**6**). In a 50 mL flask, **5** (75.3 mg, 74.3 μmol) was dissolved in acetic acid (1.6 mL), and then TFA (0.8 mL) and 5% aqueous H_2_SO_4_ (0.4 mL) were added to the solution. The mixture was refluxed at 100 ^○^C for 1.5 h. Reaction mixture was transferred to a 500 mL beaker and neutralized with saturated NaHCO_3_ aqueous solution. The product was extracted with chloroform (×5). Organic layer was washed with water (×3) and brine (×1), and dried over anhydrous Na_2_SO_4_. The solvent was evaporated to dryness, giving **6** as a purple solid (77.6 mg, quantitative). ^1^H-NMR (300 MHz, CDCl_3_, observed as atropisomers) *δ* (ppm) 10.34, 10.33 (each s, 1H, *-CHO*), 9.57 (d, *J* = 4.8 Hz, 2H, β), 9.52 (d, *J* = 4.8 Hz, 2H, β), 8.78 (d, *J* = 4.8 Hz, 4H, β), 8.73, 8.64 (each s, 1H, *Ph_2_*), 8.49, 8.41 (each d, 1H, *J* = 7.5 Hz, Ph_6_), 8.37, 8.35 (each d, partially overlapped each other, *Ph_4_*), 7.99–7.45 (m, 1H, *Ph_5_*), 7.75 (s, 2H, *Im_5_, Im_4_*), 5.09 (t, *J* = 7.2 Hz, 4H, *TEG*), 4.41 (dd, *J* = 2.4 Hz, 2H, *propargyl CH_2_*), 3.88 (s, 4H, *TEG*), 3.82–3.49 (m, 28H, *TEG*), 3.33 (s, 6H, m, *TEG*), 2.81–2.73 (m, 4H, *TEG*), 2.25 (d, *J* = 2.4 Hz, 1H, *propargyl CH*), −2.69 (s, 2H, *H_inner_*); ^13^C-NMR (100 MHz, CDCl_3_)*δ* (ppm) 192.76, 148.32, 143.68, 139.66, 134.98, 134.93, 135.09, 134.88, 131.85, 130.80, 129.64, 128.37, 128.96, 129.14, 128.89, 127.57, 127.54, 119.97, 119.86, 118.84, 103.70, 77.41, 74.50, 72.03, 70.90, 70.87, 70.83, 70.66, 70.46, 70.03, 59.13, 37.82, 37.19, 31.31.

*GablePorphyrin (GP 1Fb)*. A 300 mL three-necked flask was charged with **6** (400 mg, 4.31 × 10^−4^ mol), dipyrromethane **4** (463 mg, 1.32 × 10^−3^ mol), and chloroform (123 mL). After the solution was degassed by bubbling with Ar gas for 15 min, TFA (165 μL, 2.16 × 10^−3^ mol) diluted with chloroform was added, and the mixture was stirred under dark. After 3.5 h, 1-(prop-2-yn-1-yl)-1*H*-imidazole-2-carbaldehyde (**2**) (65.2 mg, 4.86 × 10^−4^ mol) was added. After 14.5 h, further 1-(prop-2-yn-1-yl)-1*H*-imidazole-2-carbaldehyde (**2**) (65.2 mg, 4.86 × 10^−4^ mol) was added. The reaction progress was monitored by MALDI-TOF mass. After 2 h, *p*-chloranil (583 mg, 2.37 × 10^−3^ mol) was added, and the mixture was stirred for 4.5 h. The mixture was neutralized with triethylamine (330 μL, 2.37 × 10^−3^ mol) and the solvent was evaporated to dryness. The black residue was purified by an alumina column (φ 5 cm × 12 cm, chloroform:acetone = 1:1). The red band was collected, and the fraction was concentrated to give a purple solid (392 mg). The GPC analysis showed that the solid contained the byproducts, tris-porphyrin and bis-imidazolyl-porphyrin **8**, along with ca 29% of **GP1Fb**. The mixture was purified further using preparative GPC system to afford **GP1Fb** as as a purple solid (130 mg, 17.5%). MALDI-TOF MS (matrix: DCTB) found *m*/*z* ([M + H]^+^) 1741.8659, calcd for [C_98_H_118_N_12_O_16_ + H]^+^ 1741.8686; ^1^H-NMR (400 MHz, CDCl_3_, observed as atropisomers) *δ* (ppm) 9.67 (d, *J* = 4 Hz, 4H, β), 9.57 (d, *J* = 4 Hz, 4H, β), 9.40 (d, *J* = 4 Hz, 4H, β), 9.08 (dd, *J* = 5.2 Hz, *J* = 4 Hz, 1H, *Ph_2_*), 8.80 (d, *J* = 4 Hz, 4H, β), 8.69, 8.62, 8.58 (d, *J* = 8 Hz, *J* = 4 Hz, 4H, *Ph_4_, Ph_6_*), 8.18 (m, 1H, *Ph_5_*), 7.72, 7.73 (s, 4H, *Im_4_, Im_5_*), 5.13 (t, *J* = 4 Hz, 8H, *TEG*), 3.39 (d, *J* = 20 Hz, 4H, *propargyl CH_2_*), 3.69 (m, 56H, *TEG*), 3.28 (s,12H, *TEG-CH_3_*), 2.81 (m, 8H, *TEG*), 2.25 (d, *J* = 20 Hz, 2H, *propargyl CH*), −2.59 (s, 4H, *H_inner_*); ^13^C-NMR (100 MHz, CDCl_3_)*δ* (ppm) 148.40, 140.98, 140.92, 139.92, 139.81, 134.08-133.96, 132.27, 130.66, 129.60, 128.28, 128.92, 124.97, 120.58, 119.82, 119.75, 103.35, 74.42, 74.39, 72.01, 70.91, 70.86, 70.83, 70.65, 70.48, 70.14, 37.84, 31.39, 59.09, 37.17.

*GablePorphyrin-Zn (GP1)*. **GP 1Fb** (130 mg, 7.56 × 10^−5^ mol) was dissolved in chloroform (6 mL) in a 25 mL flask. A saturated methanol solution of Zn(OAc)_2_ (100 mg, 5.45 × 10^−4^ mol) was added to the solution, and the mixture was stirred for 10 h at rt in the dark. The resulting solution was washed with water (×3) and passed through Phase Separator paper (Whatman). The solvent was evaporated to dryness, giving **GP1** as a purple solid (132 mg, 95%). HPLC-PDA (System 1, flow rate 1.0 mL/min) 21.6 min; MALDI-TOF-MS (matrix: DCTB) found *m*/*z* [M]^+^ 1865.6761, calcd for [C_98_H_114_N_12_O_16_Zn_2_]^+^ 1865.6956.

*PEG8-5Zn*. To an acetonitrile solution (1 mL) containing **5Zn** (4.3 mg, 4.0 × 10^−6^ mol) and PEG8-N_3_ (**10**, 10 mg, 2.4 × 10^−5^ mol) in a 10 mL flask was added CuI (~1 mg). The mixture was stirred for 8 days at rt in the dark under Ar atmosphere. Disappearance of the starting materials was confirmed with TLC. The mixture was passed through a Celite pad with methanol. The filtrate was evaporated to dryness, and the residue was purified with a silica gel column (φ1 cm × 9 cm, chloroform:methanol = 10:1). Fractions containing target compound was concentrated to dryness, and residue was reprecipitated from chloroform and hexane, giving a purple solid (4.8 mg, 81%). MALDI-TOF-MS (matrix: DCTB) found *m*/*z* [M]^+^ 1483.6199, calcd for [C_75_H_105_N_9_O_18_Zn]^+^ 1483.6869; ^1^H-NMR (500 MHz, CDCl_3_) *δ* (ppm) 9.61 (d, *J* = 2.7 Hz, 4H, β), 9.03 (d, *J* = 2.4 Hz, 4H, β), 8.93 (d, *J* = 2.7 Hz, 4H, β), 8.76 (s, 1H, *Ph_2_*), 8.63 (d, *J* = 3.9 Hz, 1H, *Ph_6_*), 8.24 (s, 1H, *Ph_2_*), 8.12 (d, *J* = 4.2 Hz, 1H, *Ph_6_*), 8.08 (t, *J* = 4.5 Hz, 2H, *Ph_4_*), 7.98 (t, *J* = 4.5 Hz, 1H, *Ph_5_*), 7.82 (t, *J* = 4.5 Hz, 1H, *Ph_5_*), 5.93, 7.92 (each s, 2H, *acetal-CH*), 5.87, 5.83 (each s, 2H, *Im_5_*), 5.26 (d, *J* = 2.4 Hz, 4H, *β*), 5.21–5.10 (m, 8H, *TEG*), 4.30–4.27 (m, 2H, *TEG*), 4.10–2.95 (m, *TEG*, *triazole, trazole-CH_2_*), 2.71–2.68 (m, 2H, *TEG*), 2.33–2.87 (m, 4H, *TEG*), 2.12–2.05 (m, 2H, *Im_4_,*), 1.53, 1.39 (each s, 6H, *acetal- CH_3_*), 0.95, 0.84 (each s, 6H, *acetal-CH_3_*), 1.70–1.12 (m, *TEG*).

*Py-5Zn*. In a 10 mL flask, Py-N_3[33]_ (**11**, 10 mg, 2.4 × 10^−5^ mol), **5Zn** (5.2 mg, 4.6 × 10^−6^ mol), and CuI (~1 mg) were dissolved in chloroform (ca. 1 mL) and 2,6-lutidine (a few drops). The mixture was stirred for 3 days at rt in the dark under Ar atmosphere. The target compound **Py-5Zn** was confirmed with MALDI-TOF MS. The mixture was washed with 0.5 M ethylenediaminetetraacetic acid (EDTA) aqueous solution (pH 8) and water subsequently, and the organic layer was passed through Phase Separator paper (Whatman). The filtrate was evaporated to dryness and the residue was purified with a silica gel column (φ1 cm × 20 cm, chloroform → chloroform:methanol = 10:1). The green band eluted with chloroform:methanol = 10:1 was collected, and the fraction was concentrated, giving a purple solid (5.8 mg, 95%). MALDI-TOF-MS (matrix: dithranol) found *m*/*z* [M + Na]^+^ 1354.5297, calcd for [C_55_H_68_N_6_O_10_ + Na]^+^ 1354.5296; ^1^H-NMR (400 MHz, CDCl_3_) *δ* (ppm) 9.68(d, *J* = 4.4 Hz, 2H, β), 9.15 (d, *J* = 4.8 Hz, 2H, β), 8.89 (s, 0.5H, Ph_2_), 8.78 (d, *J* = 4.8 Hz, 2H, β), 8.75 (d, *J* = 7.6 Hz, 0.5H, *Ph_6_*), 8.38 (s, 0.5H, *Ph_2_*), 8.20 (d, *J* = 7.2 Hz, 0.5H, *Ph_6_*), 8.12 (d, *J* = 7.6 Hz, 1H, *Ph_4_*), 8.04 (t, *J* = 7.6 Hz, 0.5H, *Ph_5_*), 7.86-7.82 (m, 1H, *Ph_5_*), 7.84-7.82 (m, 1.5H, *Py*), 7.71–7.62 (m, 2.5H, *Py*), 7.47 (d, *J* = 7.2 Hz, 1H, *Py*), 7.36–7.31 (m, 1H, *Py*), 6.31–6.14 (m, 2H, *Py*), 5.98, 5.71 (each s, 2H, *acetal-CH*), 5.86–5.78 (m, 1H, *Im_5_, Py*), 5.21 (dd, *J* = 4.4 Hz, 2H, β), 5.22–5.08 (m, 4H, *TEG*), 4.97–4.82 (m, 1H, *Py*), 4.08–3.49 (m, *TEG*), 3.31 (s, 6H, *TEG-CH_3_*), 3.23 (m, 4H, *triazol- CH_2_*), 3.07–3.02 (m, *TEG*), 2.14–2.20 (m, 1H, *Im_4_*), 2.09, 2.07, 2.04 (each s, 1H, *triazole*), 1.56, 1.34 (each s, 3H, *acetal-CH_3_*), 0.98, 0.82 (each s, 3H, *acetal-CH_3_*).

*F9-5Zn*. In a 10 mL flask, CuI (~1 mg) and **5Zn** (7.2 mg, 6.7 × 10^−6^ mol) were dissolved in chloroform (ca. 1 mL) and 2,6-lutidine (three drops). F9-N_3[34]_ (**13**, 19 mg, 6.7 × 10^−5^ mol) was added in the dark, and the mixture was stirred for 3 days at rt in the dark under Ar atmosphere. Disappearance of the starting materials and formation of the target compound were confirmed with MALDI-TOF MS. The mixture was washed with 0.5 M EDTA aqueous solution (pH 8) and water subsequently, and the organic layer was passed through Phase Separator paper (Whatman). The filtrate was evaporated to dryness and the residue was purified with a silica gel column (φ1 cm × 20 cm, ethyl acetate → chloroform:methanol = 5:1). The green band was collected, and the fraction was concentrated, giving **F9-5Zn** as a purple solid (8.1 mg, 88%). MALDI-TOF-MS (matrix: DCTB) found *m*/*z* [M + Na]^+^ 1386.4079, calcd for [C_64_H_74_N_9_O_10_ + Na]^+^ 1386.4604. ^1^H-NMR (400 MHz, CDCl_3_) *δ* (ppm) 9.63 (d, *J* = 4.4 Hz, 4H, β), 9.05 (d, *J* = 4.4 Hz, 4H, β), 8.87 (d, *J* = 4.4 Hz, 4H, β), 8.76 (s, 1H, *Ph_2_*), 8.62 (d, *J* = 7.6 Hz, 1H, *Ph_6_*), 8.27 (s, 1H, *Ph_2_*), 8.13 (d, *J* = 7.2 Hz, 1H, *Ph_6_*), 8.08 (t, *J* = 7.6 Hz, 2H, *Ph_4_*), 7.98 (t, *J* = 7.6 Hz, 1H, *Ph_5_*), 7.81 (t, *J* = 7.6 Hz, 1H, *Ph_5_*), 5.92, 5.71 (each s, 2H, *acetal-CH*), 5.84, 5.81 (each s, 2H, *Im_5_*), 5.26 (t, *J* = 4.4 Hz, 4H, β), 5.23-5.11 (m, *J* = 6.6 Hz, 8H, *TEG*), 4.04-3.51 (m, 56H, *TEG*), 3.33 (s,12H, *TEG-CH_3_*), 3.08-2.98 (m, 8H, *TEG*), 3.22 (t, *J* = 4.8 Hz, 4H, *Fluorine-H_1_*), 2.17, 2.12 (each s, 2H, *Im_4_*), 1.81-1.65 (m, 4H, *Fluorine-H_2_*), 1.52, 1.37 (each s, 6H, *acetal-CH_3_*), 0.95, 0.83 (each s, 6H, *acetal-CH_3_*); ^19^F-NMR (376 MHz, CDCl_3_) *δ*(ppm) −80.99 (3F, CF_3_), −114.79 (2F, CF_2_*α*), −124.60 (2F, CF_2_*β*), -126.11 (2F, CF_2_*γ*).

*PEG8-GP1*. According to a similar procedure to give **Py-5Zn** and **F9-5Zn**, **PEG8-GP1** (14.8 mg) was prepared from PEG8-N_3_ (18 mg, 4.4 × 10^−5^ mol), **GP1** (12 mg, 6.3 × 10^−6^ mol), CuI (8 mg, 4.0 × 10^−5^ mol), and 2,6-lutidine (12 mg, 1.0 × 10^−4^ mol) in a 88% yield as a purple solid. HPLC-PDA (System 1, flow rate 1.0 mL/min) 20.4 min; MALDI-TOF-MS (matrix: DCTB) found *m*/*z* [M + Na]^+^ 2684.1042, calcd for [C_132_H_184_N_18_O_32_Zn_2_ + Na]^+^ 2684.1805.

*Py-GP1*. According to a similar procedure to give **Py-5Zn** and **F9-5Zn**, **Py-GP1** (13 mg) was prepared from Py-N_3_ (7 mg, 2.7 × 10^−5^ mol) and **GP1** (10 mg, 5.4 × 10^−6^ mol) in a quantitative yield. HPLC-PDA (System 1, flow rate 1.0 mL/min) 21.4 min; MALDI-TOF-MS (matrix: DCTB) found *m*/*z* [M + Na]^+^ 2379.8734, calcd for [C_132_H_136_N_18_O_16_Zn_2_Na]^+^ 2379.8862.

*BA-GP1*. According to a similar procedure to give **Py-5Zn** and **F9-5Zn**, **BA-GP1** (5.5 mg) was prepared from BA-N_3_ (45 mg, 1.5 × 10^−5^ mol) and **GP1** (2.5 mg, 1.4 × 10^−6^ mol) in a quantitative yield. HPLC-PDA (System 1, flow rate 1.0 mL/min) 20.8 min; MALDI-TOF-MS (matrix: DCTB) found *m*/*z* [M + Na]^+^ 2456.2474, calcd for [C_134_H_188_N_18_O_16_Zn_2_Na]^+^ 2456.2931.

*F9-GP1*. According to a similar procedure to give **Py-5Zn** and **F9-5Zn**, **F9-GP1** (5.5 mg) was prepared from **GP1** (3.5 mg) and F9-N_3_ (**13**, 16.8 mg, 5.8 × 10^−5^ mol) in a quantitative yield. HPLC-PDA (System 1, flow rate 1.0 mL/min) 21.0 min; MALDI-TOF-MS (matrix: DCTB) found *m*/*z* [M + Na]^+^ 2443.6951, calcd for [C_110_H_122_F_18_N_18_O_16_Zn_2_Na]^+^ 2443.7479.

## 4. Conclusions

Here, we have synthesized new Gable porphyrin **GP1** with hydrophilic TEG side chains and prop-2-yne groups as a scaffold to introduce various functional groups. Four types of functional groups, namely, hydrophilic oligoether group (PEG8), pyrene derivative (PY), BA, and a nonafluorinated moiety (F9), were introduced into **GP1** in high yields to give **PEG8-GP1**, **PY-GP1**, **BA-GP1**, and **F9-GP1**, respectively. Formation of heterogeneous macrocyclic pentamers and hexamers composed of **PEG8-GP1** and **PY-GP1**, and **PEG8-GP1** and **F9-GP1** was demonstrated by recombination of the two components. These results indicate that different types of substituents can be mixed through the macrocycles, and promise that numerous kinds of macrocycles having various functional groups can be provided when the kinds of GP components increase. Under equilibrium conditions, components of the macrocycles are exchanged. These features will be beneficial for DCC, and the present system using **GP1** is a potential candidate to provide a DCL of multitopic probes to discover specific interactions between ligands and biomaterials.

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
