# Peer review of "A Candidate for Multitopic Probes for Ligand Discovery in Dynamic Combinatorial Chemistry"

_molecules, 2019, doi:10.3390/molecules24112166_

Round 1

Reviewer 1 Report

This is an interesting manuscript about the application of gable porphyrins to create DCC libraries by Zn-imidazole coordination. Citcumstances are given to amke the macrocycles dynamic and extensive analysis is done. The authors have published already much similar research, now they make diverse gable porphyrin elements by click reaction.  I would prefer to use the term "click reaction" or "CuAAC" instead of Huisgen reaction which actually is a 1,3-dipolar cycloaddition operation via a different mechanism.

At the moment the link to medicinal relevance of these macrocycles seems rather weak, the study is more of a fundamental importance .

Author Response

The authors gratefully thank you for your valuable comments.

A term “Huisgen reaction” was replaced to copper(I)-catalyzed alkyne-azide cycloaddition or CuAAC.

As you suggested, the most important results are synthesis and characterization of new compounds, but not DCC. We emphasize these results, and also reduced unnecessary descriptions toward DCC.

As suggested by another reviewer, “2. Results” and “3. Discussion” parts were combined as “2. Results and discussion” in the revised document to be more readable and understandable.

Please find an attached document in which revised points are listed.

Reviewer 2 Report

In the paper, the authors studied the self assembly of zinc porphyrin components bearing varied pending groups like pyrene, PEG, perfluorinated and alkyl chains. These assemblies were investigated by NMR, mass, and UV-vis spectroscopies and GPC. Pentamers and hexamers were formed and their recombination was highlighted by previously mentioned analyses. The paper deserves publication in Molecules after minor corrections.

General remark: a title should be given to all schemes describing, for instance, name of reaction, compounds code etc.

P2 line 77 the authors could give more information regarding the “special pairs”.

P7 line 171 scheme 1: the ratio of compounds 5, 7, 8 could be given.

P9 line 200 figure 8a & b: “beta” should be explained for clarity (scheme 4?). It could also help the reader to show Im 4 & 5 on fig 8a before addition of Zn.  

P11 line 239 scheme 5: title is unclear.

P12 figure 10 proCH2 and proCH should be given in scheme 6.

P15 line 305 “in the absence of MeOH?”

P16 figure 14b 448 & 446 nm: the authors should clearly indicate in the title the assignment of these species.

P17 figure 16 the authors should refer to PEG8-GP1 in the title.

P20 figure 19 the authors should refer to Py-GP1 & PEG8-GP1 in the title.

P24 line 493 is there a possibility of pi pi stacking between the pyrene group and zinc porphyrin ring?

Author Response

Answer for Reviewer 2:

The authors gratefully thank you for your kind indications and valuable comments. We revised the manuscript according to all of the comments.

General remark: a title should be given to all schemes describing, for instance, name of reaction, compounds code etc. -> OK

P2 line 77 the authors could give more information regarding the “special pairs”.

-> revised as follows:

so-called “special pair” in photosynthetic reaction center of photosynthetic purple bacteria. The slipped-cofacial dimer of the imidazolyl zinc porphyrins mimics arrangement of the bacteriochlorophylls of the special pair.

P7 line 171 scheme 1: the ratio of compounds 5, 7, 8 could be given.

-> revised in the Figure caption

P9 line 200 figure 8a & b: “beta” should be explained for clarity (scheme 4?). It could also help the reader to show Im 4 & 5 on fig 8a before addition of Zn.  OK

P11 line 239 scheme 5: title is unclear. OK

P2 figure 10 proCH2 and proCH should be given in scheme 6.  OK

P15 line 305 “in the absence of MeOH?”   OK

P16 figre 14b 448 & 446 nm: the authors should clearly indicate in the title the assignment of these species.   OK

P17 figure 16 the authors should refer to PEG8-GP1 in the title.   OK

P20 figure 19 the authors should refer to Py-GP1 & PEG8-GP1 in the title.   OK

P24 line 493 is there a possibility of pi pi stacking between the pyrene group and zinc porphyrin ring? -> possible. Added as well as PY and zinc porphyrin moieties

Reviewer 3 Report

This manuscript by Satake and co-workers is concerned with the use of imidazole zinc porphyrins in the field of dynamic combinatorial chemistry. The authors prepared a series of imidazole zinc porphyrin derivatives decorated with perfluoroalkyl chains, pyrene units, bulky highly branched alkyl chains and polyethyleneglicol fragments and studied their ability to generate dynamic libraries through different techniques.

At the present stage, the most interesting results of the manuscript are the syntheses and characterizations of the new species, which should deserve more emphasis in the text. Concerning the DCC study involving the above species, frankly speaking, I have found its description very hard-to-digest. In other words, the manuscript in the present form resemble more a laboratory report than a scientific article. It is my opinion that the manuscript should be re-written, particularly in the discussion which is a sort of “shopping list”, in order to be more readable and understandable.

Most part of the scientific results is sound, but not the way such results are presented. My suggestion is to reject the manuscript and to ask the authors to resubmit the work that should be more concise with a proper discussion of the results.

Author Response

Answers for Reviewer 3:

The authors gratefully thank you for your honest comments. In the 1st document, we divided “2. Results” and “3. Discussion” according to the document template prepared by MDPI. However, this style was not suitable for our present paper as you suggested. Therefore, they were combined as “2. Results and discussion” in the revised document. We believe that the revised manuscript becomes more readable and understandable. Also, we agree that the 1st document bored you with excess use of term “DCC”. In the revised document, we emphasize results concerning with synthesis and characterization of new species, and also reduced unnecessary descriptions toward DCC. In an attached document, we listed the revised parts in detail.

Round 2

Reviewer 3 Report

I am satisfied with the changes done by the authors